# OpenReview forum: "WebGauntlet: Measuring Instruction Following and Robustness for Web Agents"
_ICLR.cc/2025/Workshop/BuildingTrust — Submitted to BuildingTrust_

### Official Review · Reviewer_dnb9 · 2025-03-01
**Potentially Useful Web LM Agent Benchmark with Limited Evaluation and Presentation Issues**

**Rating:** 5
**Confidence:** 3

**Review:**

Summary:
The authors propose a benchmark for evaluating LM agents in adversarial web e-commerce tasks. They construct an e-commerce site with a scrollable product grid, detailed product pages, a shopping cart, and a product search engine. The benchmark introduces three threat patterns—benign, human-specific, and agent-specific—which manifest as pop-ups, banners, ad slots, etc. The authors evaluate two LMs, GPT-4o and Claude-3.5 Sonnet, finding both susceptible to different threat patterns, with attack success rates (ASR) reaching 98.92%.

Originality and Significance: The work builds on web LM agent research and introduces an adversarial evaluation benchmark for web e-commerce tasks, making a timely contribution to the safe deployment of web LM agents.

Pros:

- Evaluating LM-based web agents in adversarial settings is timely and important
- The benchmark effectively highlights vulnerabilities in VLM web agents, demonstrating high ASR (up to 98.92%)
- A variety of attack types and placements are considered

Cons:

- The paper considers a limited model pool with only gpt-4o and claude-3.5-sonnet
- No experiments exploring the impact of prompting techniques or safety-prompting on agent performance, potentially undermining LM agent performance
- Statistics on the test cases are lacking, e.g. how many examples are evaluated for each threat model?
- The claim made in subsection "Agents quickly learn how to get by benign attacks" in section 5 is unclear and lacks references to supporting tables or results
- Presentation issues: uses inconsistent terminology (e.g., "Normal" used instead of "human-specific" in Table 3) and broken figure reference in Section 5; overall lacklustre presentation

---

### Official Review · Reviewer_Ruve · 2025-03-01
**Great engineering efforts with interesting robustness findings for Web Agents under environment-side attacks, but with major presentation issues and limited scope.**

**Rating:** 5
**Confidence:** 5

**Review:**

Pros:

1. The work demonstrates substantial engineering effort in building and hosting a reproducible sandbox environment with realistic adversarial web attacks.
2. Interesting results showing state-of-the-art agents (based on GPT-4 or Claude) are vulnerable to webpage manipulations.
3. Dividing attacks into Benign, Human, and Agent categories provides structured insights into specific failure modes

Cons:

1. presenting style
    1. Noticeable presentation issues in (e.g., Fig. 7(Blank figure), Appendix C (out of bounds), etc.)
    2. Overall writing resembles a technical report rather than a formal scientific publication (e.g., missing rigorous definitions or more formalized exposition).
2. Limited domain of only e-commerce
    1. The study focuses exclusively on an e-commerce scenario, limiting broader generalization.
    2. Most evaluations appear restricted to single-site settings, leaving multi-domain or multi-site unexplored
3. Single Agentic framework (WebVoyager) tested, would be a plus to test on more recent open sourced framework such as OpenHands(https://arxiv.org/abs/2407.16741), and specialized web agent such as ScribeAgent (https://arxiv.org/abs/2411.15004)
4. Different frameworks leverage distinct observation formats (screenshots vs. DOM trees). Expanding tests to agents with other observation strategies would help validate whether findings hold universally.

Overall, the paper requires some revision, although it develops and tests with several attacks, which is valuable, it has a really limited scope and systems tested. Additionally, the paper has several presenting issues that should be modified.

---

### Official Review · Reviewer_kUok · 2025-03-02
**Review for the WEBGAUNTLET Paper (Measuring Instruction-Following for Web AI Agents)**

**Rating:** 2
**Confidence:** 4

**Review:**

# Summary
This paper introduces WEBGAUNTLET, a novel benchmark that assesses the safety and resilience of language model (LM) agents in practical online e-commerce environments. The environment simulates typical e-commerce websites with integrated adversarial content, allowing researchers to analyze the degree to which agents follow instructions and are resistant to several online attacks. The benchmark integrates a diverse threat model with different types of attacks, such as redirections, data scraping, and system notifications, placed in many different positions on the website.

Authors' experiments with WEBGAUNTLET demonstrate that current LM agents struggle to navigate through even simple adversarial material, with highly effective attacks. Comparing this with the human baseline, the human baseline does not have a problem completing the tasks, highlighting the vast gap between human and agent performance in these environments. With the release of the WEBGAUNTLET environment, authors wish to encourage more work on increasing the safety and robustness of web agents.

# Pros and Cons
## Pros
1. **Results:** Some of the results seem interesting. However, a clear attack model is missing, making it hard to interpret the impact of the paper.
2. **Motivation:** The overall motivation behind the paper is well-justified. However, there is a clear lack of motivation (or lack of expressed motivation) behind many design choices made for the construction and evaluation of the benchmark.

## Cons
1. **Quality and Completeness:** The paper seems to be written in a rushed manner with some figures missing (line 334) and many concepts and definitions are not well-explained. An arbitrary decision-making process seems to have been involved for the different parts of the benchmark generation, including for using "agent-specific" instructions for the attacks (or jailbreaks if that's what they mean), the overall categories of attacks (Benign, Human, Agent-Specific), and the "operational modes" (single-mode vs multi-mode). There seems to be a major lack of clarity for these concepts in the paper. There are also lots of grammatical and dictation errors throughout the paper. In terms of the completeness, there seems to be terms (such as "the randomization algorithm") used in specific contexts before which they are not well-explained at all.
2. **Scope:** The scope of the benchmark seems too limited to be practically useful for real-world evaluations of the robustness of web agents. It only consists of a single simulation direction (online e-commerce simulations), making it bound to a very specific context. And even within this context, the structure of the simulation platform designed by the authors seems to be not very flexible. It is particularly unclear what safety/security properties are being evaluated with each of the tests/samples and the evaluation directions seem too broad.
3. **Data Generation:** It is very unclear how the data is generated for the benchmark and how it could be generated in a scaled manner to further generalize this approach to the generation of more complex or domain-specific benchmarks.
4. **Novelty:** I see that defining this new benchmark with "simple" tasks that the authors claim the agents fail on could be considered fairly novel, but I think this idea has to be much better developed into actually building a robust benchmark that can useful for the community evaluating the robustness of web agents in a reliable manner. The over-generalizations and lack of systematic design in the benchmark generation process make it hard to see the reliability aspect.

# Minor/Major Errors
1. "agent-specific" should be used instead of "agent specific" on line 202.
2. "deployed" should probably be used instead of "deploy" on line 232? This part of the paper, i.e. the "Operational Modes" section, is also very unclear.
3. "successfully" should be used instead of "successful" on line 255.
4. Many grammatical errors on line 257.
5. **Figure missing on line 334.**


# Questions
1. Shouldn't there be citations for the defined metrics? Do the authors claim that the metrics are completely novel?

---

### Decision · Program_Chairs · 2025-03-04

Reject